# The effectiveness of interactive mobile health technologies in improving antenatal care service utilization in Dodoma region, Tanzania: A quasi—Experimental study

**Theresia J. Masoi**[1], **Stephen M. Kibusi**[2], **Deogratius Bintabara**[3]*, **Athanase Lilungulu**[4]

**1** Department of Clinical Nursing, the University of Dodoma, Dodoma, Tanzania, **2** Department of Public Health and Community Nursing, the University of Dodoma, Dodoma, Tanzania, **3** Department of Community Medicine, the University of Dodoma, Dodoma, Tanzania, **4** Department of Obstetrics and Gynecology, the University of Dodoma, Dodoma, Tanzania

* bintabaradeo@gmail.com

**Data Availability Statement:** The dataset for this research cannot be made publicly available for ethical purposes. Public availability of this data

## Abstract

Antenatal care (ANC) provides a platform for important health care during pregnancy, including health promotion, screening, diagnosis and disease prevention. Timely and appropriate utilization of antenatal care can prevent complications as well as ensure optimal maternal and newborn health care. This study assessed the effectiveness of interactive (two way communication) mobile health technologies during antenatal period to improve maternal and newborn service utilization in Dodoma region, Tanzania. Using quasi-experimental design, participants were randomly selected to achieve a sample size of 450 pregnant women (Intervention = 150 and Control = 300) in Dodoma city from January to November, 2018. Interventions were matched to controls by gravidity, education level and gestational age at a ratio of 1 to 2. The intervention group received health education messages through their mobile phones, while the control group continued with standard antenatal care services offered in local clinics. Pregnant women were followed from their initial visit to the point of delivery. The Chi-square test was used to establish the association and regression analysis were used to test the effect of the intervention. The median age of participants was found to be 25 years that ranged from 16 to 41 years. Generally, 77.3 percent of participants in the intervention group utilized adequate (i. ANC care provided by skilled health personnel, ii. Sufficient number of ANC visits (4 or more visits during pregnancy), iii. Appropriate ANC contents provided (visits included at least 13 out of 15 of the recommended basic care procedures or contents) ANC services compared to 57.7 percent in the control group. Interactive mobile health technology system was observed to be effective on improving antenatal care service utilization (AOR = 2.164, P<0.05, 95% CI = 1.351–3.466) compared to conventional antenatal care health education given in local health facilities. Use of interactive mobile health technologies during antenatal period has the potential of improving access to information and antenatal care service utilization in the study setting.

**Trial Registration**: PACTR202008834066796 "Retrospectively registered".

might compromise patient privacy. However, it can be accessed upon reasonable request from the Directorate of Research Publication and Consultancy, University of Dodoma, P.O. Box 259, Dodoma, Tanzania (drpc@udom.ac.tz).

**Funding:** The authors received no specific funding for this work.

**Competing interests:** The authors have declared that no competing interests exist.

## Author summary

Increasing the utilization of maternal health care services is an important strategy to reduce preventable maternal morbidity and mortality. Each year, roughly a third of global maternal deaths are due to inadequate care during pregnancy. Antenatal care visits are ideal time to advice women and their families on essential pregnancy care and develop a birth plan and complications readiness. These approaches improve outcomes for women and may also reduce stillbirths and neonatal deaths. Improving maternal and neonatal health outcomes involves the provision and uptake of antenatal services that are timely (first visit during the first three months of pregnancy), sufficient (at least four antenatal visits) and adequate with appropriate contents. The use of mobile health interventions such interactive messaging system can provide behavioural support and health education needs to pregnant women.

This study has demonstrated the importance of interactive mobile health technology in improving antenatal care service utilization compared to conventional method. The interactive mobile method raised awareness and served as the best alternative tool to provide health education among pregnant women. Consistent use of SMS technology to disseminate health information is a promising approach to improve monitoring of pregnant women and increase maternal health care service utilization.

## Introduction

Health care services during pregnancy, childbirth and after delivery are important for survival and well-being of both the mother and the newborn. Identification and management of antenatal risk factors at early stages are important for positive maternal and newborn outcomes. Timely and appropriate utilization of antenatal care can prevent complications as well as ensure better maternal and newborn health care [1]. The World Health Organization (WHO) has recommended the following three core health sector strategies for reducing maternal and early neonatal deaths: comprehensive reproductive health care; skilled care for all pregnant women, especially during delivery; and emergency obstetric care for all women as well as infants with life-threatening complications [2].

Antenatal care (ANC) service being one of the packages of reproductive health care, provides a platform for important health care during pregnancy including health promotion, screening, diagnosis and disease prevention [2]. In addition, antenatal care provides an opportunity to communicate and support women, families as well as communities [3].

Despite various efforts to improve maternal health, maternal and neonatal mortality rates have remained unacceptably high in many developing countries, including Tanzania [4].

In due regard, increasing utilization of maternal health care services is an important strategy to reduce preventable maternal morbidity and mortality. For each year, roughly a third of global maternal deaths are due to inadequate care during pregnancy. Thus, antenatal care visits are an ideal time to advice women and their families on essential pregnancy care so as to develop a birth preparedness and complications readiness plan. These approaches improve outcomes for women and also, may reduce stillbirths as well as neonatal deaths. Improving maternal and neonatal health outcomes involves provision as well as uptake of antenatal services that are timely (first visit during the first three months of pregnancy), sufficient (at least four antenatal visits) as per previous guideline, currently Tanzania has adopted a new WHO guideline which emphasis that pregnant women should have at least eight or more contact [5] and adequate with appropriate content [2].

Results from a study conducted by World Health Organization on antenatal care led to estimate that worldwide, only 70 percent of pregnant women receive ANC services, whereas in industrialized countries, more than 95 percent of pregnant women receive ANC [6]. According to the 2015/2016 Demographic and Health Survey (TDHS) conducted in Tanzania, twenty-four percent of pregnant women started antenatal care in their first trimester and a bit over half (51%) had four or more ANC visits as recommended. In the 2004/05 TDHS report, it was uncovered that 94 percent of pregnant women had at least one antenatal care (ANC) visit and 62 percent of women had four or more ANC visits and the median gestational age of pregnant women in their first ANC visit was 5.4 months [7].

As part of the Sustainable Development Goals (SDGs), the United Nations set a target to reduce global maternal mortality ratio to less than 70 per 100,000 live births between the years 2016 and 2030 with no individual country exceeding a Maternal Mortality Ratio (MMR) of 140 maternal deaths per 100,000 live births. Part of SDG Number 3, Good Health and Well-Being for People, aims to achieve universal health coverage including access to essential medicines and vaccines [8]. Therefore, interventions that target to improve maternal antenatal attendance are of paramount importance in order to curb adverse pregnancy outcomes.

There is paucity of published data on use of Short Message Service, (SMS) with cell phones as a vehicle to increase health knowledge and potentially change of women's behavior during pregnancy [9]. Besides, there is little evidence regarding different types of mobile health applications that can be used in low-resource settings [10]. In this regard, this study assessed effectiveness of interactive mobile health technologies in improving antenatal care service utilization in Dodoma region, Tanzania.

## Materials and Methods

### Study design

The study was a quasi-experimental undertaking with control group. The intervention group was enrolled in an interactive system and received health education messages through their mobile phones pertaining to their pregnancy, whereas control group went on with the normal standard ANC services offered at ANC local clinics.

### Study setting

This study was carried out at Dodoma city for both the intervention group and control group from January to November, 2018. Dodoma city is one of Tanzania's 30 administrative regions as well as a capital city of the country. It lies centrally in the eastern-central part of the country about 480 kilometers due east from the coast. Dodoma Urban District is one of seven districts of Dodoma region. It is bordered to the west by the Bahi district, and to the east by Chamwino district. According to the 2012 Tanzania national census, the population of Dodoma urban district was 410,956 and covers 2,576 kilometer squares [11]. Dodoma is one of regions with the highest maternal mortality rates in Tanzania such that in 2012, Dodoma ranked the ninth high burdened region with a maternal mortality rate of 512 per 100,000 live births [12].

In Dodoma city, there are two major public health facilities, namely, Makole Health Center, which serves as the main antenatal care facility and Dodoma Regional Referral Hospital that caters for all deliveries of high-risk mothers. In this study, the interactive mobile health was developed and pregnant mothers attending antenatal care services at Makole Health Center as well as Chamwino Dispensary were assigned as the intervention group, followed as well as received text messages regarding their pregnancy. Control groups were taken from other facilities offering Reproductive and Child Health (RCH) and delivery services in Dodoma city. They were not selected for the intervention and they included the following public health

facilities: Hombolo Health Center, Mkonze Health Centre and one private health facility, Saint Gemma Hospital.

## Sample and sample size

All pregnant women who started their first antenatal visit below the first twenty weeks formed the study population. Control groups involved pregnant women who used the current antenatal care assessment modality as per Tanzania guidelines who also started their first visits below the first 20 weeks. Both control and intervention groups were matched by age group, education level, gravidity parity and gestation age.

The sample size for the intervention group and control group was obtained by using the formula for comparing two independent samples (intervention group against control group).

$$n = 2\{z\alpha\sqrt{[\pi0(1-\pi0)]} + Z\beta\sqrt{[\pi1(1-\pi1)]}\}/(\pi1-\pi0)2$$

Then the research team used proportion of women attending four visits or more at baseline that accounted for 51 percent and after intervention it was 63 percent as per [13] through their Quasi-experiment they carried out in Rural Uganda on Maternal Health Service utilization and Newborn care. The normal standard of 1.96 was at 95 percent confidence interval (CI) with 5 percent attrition rate. Thus, the minimum sample size was 144 plus 5 percent attrition that was equal to 150. The ratio of the intervention group to control was 1 to 2 and thus, control group participants were 300. Therefore, the total sample size in this study was 450 pregnant women for both intervention group and control group.

## Sampling procedures

To ensure adequate coverage, purposive sampling procedure was employed to identify health facilities offering antenatal and intrapartum services in Dodoma city. Participants were assigned into intervention and control arms through systematic random sampling. A sampling interval of three was employed from sampling point to recruit participants who met inclusion criteria.

## Inclusion and exclusion criteria

Pregnant women who owned phones and booked ANC at the gestation less than 20 weeks and planned their deliveries in Dodoma city, were included in the study after signing a consent form. Those who were enrolled in similar intervention programs were not eligible to participate in the study because including them could result into information contamination and bias.

## Measurements of variables

**Antenatal care service utilization.** Adequate antenatal care service utilization was measured along the following aspects:

i. ANC care provided by skilled health personnel (a nurse or a doctor);

ii. Sufficient number of ANC visits (four or more visits during pregnancy); and

iii. An appropriate ANC contents provided (included at least 13 out of 15 of the recommended basic services/procedures or contents).

Individuals scoring three of the basic components (attended by skilled health personnel, 4 or more visit during pregnancy with appropriate contents/services), were classified as received

adequate ANC, while those who did not comply with the three criteria were termed as received inadequate care. Adequate ANC care corresponded to receiving thirteen out of fifteen basic ANC services. Such method of scoring has been previously used in Mexico to measure adequacy of antenatal health care and then it was adapted as well as modified to fit the Tanzanian context [2].

Components of antenatal care services offered in Tanzania include the following:

Blood test for hemoglobin level, urine test for protein as well as sugar, obstetric examination, blood pressure measurement, maternal weight/height, rapid syphilis test, blood type and Rhesus (Rh) factor testing, tetanus toxoid vaccine and iron/folic acid supplementation. Others include anti-malaria prophylaxis and client health education and counseling as well as Human Immunodeficiency Virus/Acquired Immunodeficiency Disease (HIV/AIDS) testing and one ultrasound scan in the first 24 weeks of pregnancy [14]. Currently, all such details are recorded on the antenatal card (RCH4).

## Data collection methods

An English version questionnaire was developed and then translated into Kiswahili questionnaire with close-ended and open-ended questions was used to collect information on socio-demographic characteristics as well as antenatal care service utilization. The questionnaire was adapted from Johns Hopkins Program for International Education in Gynacology and Obstetrics (JHPIEGO), Tanzania Demographic Health Survey 2015/2016 as well as Nepal Demographic Health Survey and it was modified to fit the local context [14,15].

Seven research assistants who are medical personnel (doctor and nurses) were recruited and trained for two days on the objectives of the study, interviewing techniques and how to use the data collection tool in the field. This was important for them to be familiar with the study and give them enough experience in collecting information in the field. The research assistants were all fluent in Swahili language.

## Overview of the Intervention

The interactive mobile health system is a computerized system that ensures provision of reproductive health and communication between a pregnant woman and medical practitioners (a two-way communication). The system started working in January, 2018 whereby pregnant women who started their first antenatal visit below 20 weeks were recruited and enrolled at Makole Health Centre and Chamwino Dispensary. Participants were followed until delivery. The system tracks every visit diagnosis and assembles them to give the general health overview of a woman in each trimester of the visit. Also, the system included an SMS Module where pregnant mothers were notified with SMS (reminder) texts about the time to go for the next visit. The system generated the reminder messages automatic based on the date of the previous ANC visit and the gestation age. The pregnant women were also given health tips pertaining their pregnancy such as health education on key danger signs, birth preparedness and complications readiness. Health education tips specific on ANC service utilization covered the following aspects; the importance of completing all the required visits as per Tanzania national guidelines (old modal) which is four or more visits. They also received health education messages on blood test for hemoglobin level, blood group type and Rh factor testing, urine test for protein and glucose, obstetric ultrasound at least one in each trimester, blood pressure measurement in each visit, maternal weight gain and nutritional support, tetanus toxoid vaccine, iron/folic acid supplementation, antimalaria prophylaxis. Furthermore, received health education tips on HIV/AIDS counseling and testing as well as family planning use after delivery. Participants were able to call and send text

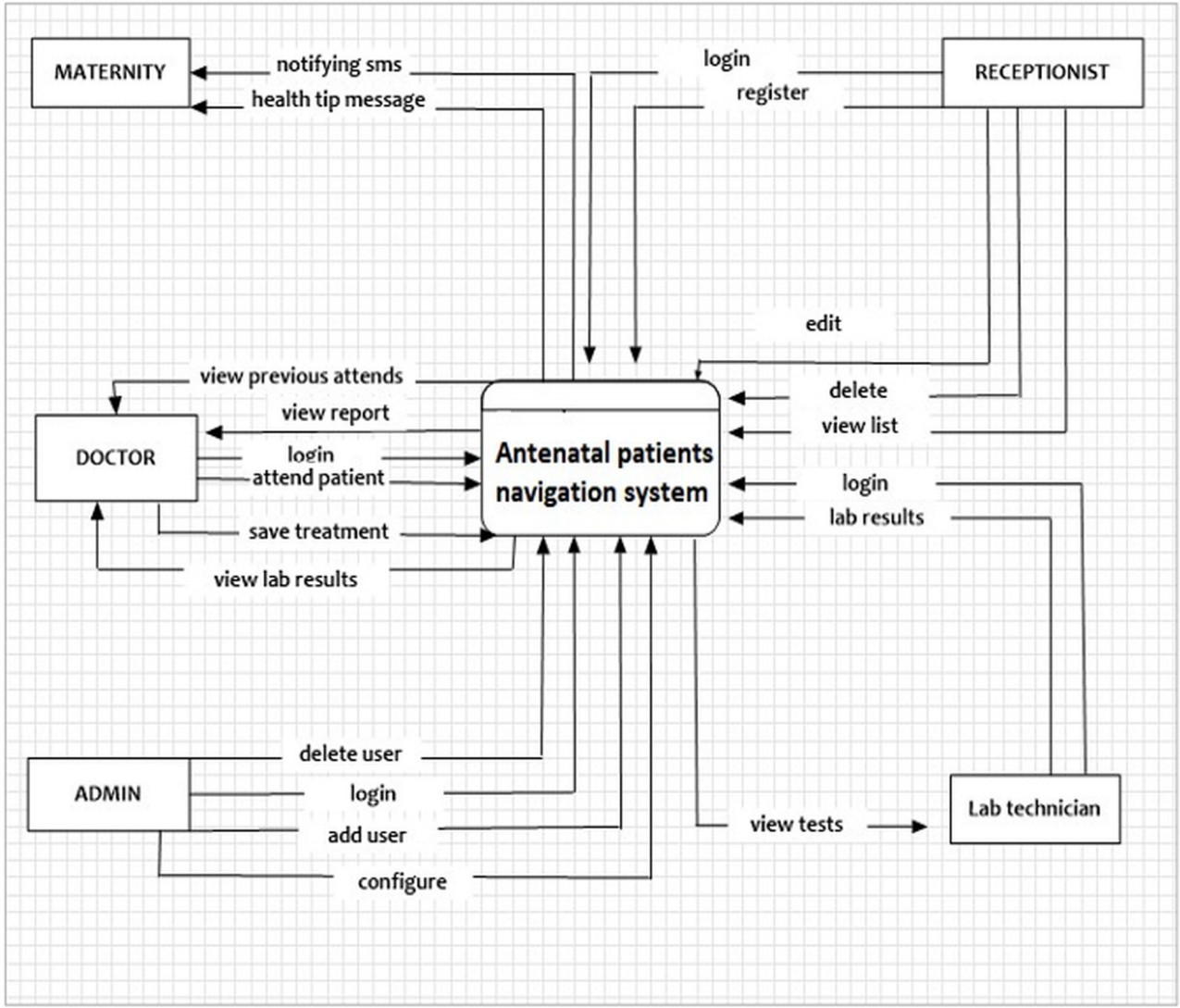

**Fig 1. Flow chart depicting flow of data/information in and outside the system.**

messages for clarification when need arose as it was a two ways communication. Pregnant women were only required to provide their mobile phone numbers and mobile phone numbers of their spouses if available. Message content was checked for standard and provided as simple SMS in the local language of Swahili. The constructed SMS did not exceed 480 words equivalent to three SMS.

The System was implemented with the back-up mechanism, whereby every day the system stored data to the backup device. This was to help in case of any crises data could be recovered on time. The system was also implemented with current technology to ensure security and prevent hackers to hack the system. The system was updated regularly to ensure any vulnerability/open door is closed. There was a password that protected web user interface, and enabled every user to access only the information required at his /her position in the system. (see Fig 1).

## Data processing and data analysis

Collected data were sorted, cleaned, arranged, coded and entered in a spreadsheet. Then such data were analysed through Statistical Package for Social Sciences (SPSS) program version 21.

Descriptive analysis was used to determine frequencies and percentages of distributions between the two arms and it was used to get the overall percentage of the ANC service utilization between the two groups. Categorical data were analyzed using Chi-square test to establish the association. Besides, Logistic regression analysis was used to establish association between the interactive mobile health system and ANC service utilization. A confidence interval of 95 percent and the margin of error of 5 percent (0.05) were used as statistical measures of significance.

## Ethical considerations

Permission to conduct this study was obtained from University of Dodoma Research Committee. Ethical research clearance and a research approval letter were obtained from the Graduate Office University of Dodoma with the approval number (Ref: UDOM /DRP/134/VOL V/23-33). Authorization to conduct the study in Dodoma Municipal and in the selected health facilities was obtained from Dodoma Urban District director and medical officer in charges. Human rights, privacy, and confidentiality were considered in this study. Research objectives, risk, and benefits of the study were explained well to the participants. Verbal and written consent were obtained from the participants and the questionnaires were answered voluntarily. Also, the control group was not denied their right to ANC. They continued receiving the normal standard ANC care available in their local facility.

# Results

## Participants' socio-demographic characteristics

Recall, a total of 450 (intervention group involved 150 pregnant/postnatal women and 300 postnatal women as control group) were recruited as well as participated in this study. The median age of respondents in the entire sample was 25 years with the range of 16 to 41 years. As shown in Table 1, the most prominent age group ranged from 20 to 34 years. There was no

**Table 1. Participants' socio-demographic characteristics (N = 450).**

| Variable | | Intervention | | Control | | Total/450 | | P-value |
|---|---|---|---|---|---|---|---|---|
| | | n | % | n | % | n | % 3.256 | |
| **Age** | <20 yrs | 21 | 14.0 | 61 | 20.3 | 82 | 18.2 | |
| | 20–34 yrs | 111 | 74.0 | 212 | 70.7 | 323 | 71.8 | 0.196 |
| | ≥35 yrs | 18 | 12.0 | 27 | 9.0 | 45 | 10.0 | |
| **Education status** | | | | | | | | |
| | Primary school | 82 | 54.7 | 182 | 60.7 | 264 | 58.7 | |
| | Secondary school | 52 | 34.7 | 95 | 31.7 | 147 | 32.7 | 0.381 |
| | College/university | 16 | 10.6 | 23 | 7.6 | 39 | 8.6 | |
| **Occupational status** | | | | | | | | |
| | Non-employed | 54 | 36.0 | 90 | 30.0 | 144 | 32.0 | |
| | Self-employed | 92 | 61.3 | 206 | 68.7 | 298 | 66.2 | 0.230 |
| | Employed | 4 | 2.7 | 4 | 1.3 | 8 | 1.8 | |
| **Marital status** | | | | | | | | |
| | Not married | 25 | 16.7 | 72 | 24.0 | 97 | 21.6 | |
| | Married | 125 | 83.3 | 228 | 76.0 | 353 | 78.4 | 0.075 |

**Table 2. Obstetric characteristics of participants N = 450.**

| Variable | | Intervention | | Control | | Total/out of 450 | | p-value |
|---|---|---|---|---|---|---|---|---|
| | | n | % | n | % | n | % | |
| **Gravidity** | 1 | 33 | 22.0 | 64 | 21.3 | 97 | 21.6 | 0.910 |
| | 2–4 | 109 | 72.7 | 217 | 72.3 | 326 | 72.4 | |
| | ≥ 5 | 8 | 5.3 | 19 | 6.4 | 27 | 6.0 | |
| **Parity** | 1 | 37 | 24.7 | 84 | 28.0 | 121 | 26.9 | 0.689 |
| | 2–4 | 107 | 71.3 | 202 | 67.3 | 309 | 68.7 | |
| | ≥ 5 | 6 | 4.0 | 14 | 4.7 | 20 | 4.4 | |
| **Gestation age at delivery in weeks** | | | | | | | | 0.289 |
| <37 weeks | | 17 | 11.3 | 38 | 12.7 | 55 | 12.2 | |
| 37–40 weeks | | 116 | 77.4 | 241 | 80.3 | 357 | 79.4 | |
| ≥ 40 weeks | | 17 | 11.3 | 21 | 7.0 | 38 | 8.4 | |
| **Gestation age at first visit in weeks** | | | | | | | | 0.106 |
| 1–12 weeks | | 56 | 37.3 | 136 | 45.3 | 192 | 42.7 | |
| 13–20 weeks | | 94 | 62.7 | 164 | 54.7 | 258 | 57.3 | |
| **Age at first pregnancy in years** | | | | | | | | 0.005 |
| <20 years | | 58 | 38.7 | 156 | 52.0 | 214 | 47.4 | |
| 20–34 years | | 90 | 60.0 | 143 | 47.7 | 234 | 52.0 | |
| ≥35 years | | 2 | 1.3 | 1 | 0.3 | 3 | 0.6 | |
| **Total number of antenatal care visits** | | | | | | | | 0.000 |
| 1–3 visits | | 14 | 9.3 | 72 | 24.0 | 86 | 19.1 | |
| ≥4 visits | | 136 | 90.7 | 228 | 76.0 | 364 | 80.9 | |

significant difference in age distribution between the two groups with p equals to 0.196). Besides, more than half (54.7%, n = 82) of the participants in the intervention group against 60.7 percent (n = 182) in the control group had primary school education level and few had college/university education (10.6%, n = 16) in the intervention group and 7.6 percent (n = 23) in the control group. Out of 450 respondents, 78.4 percent were in marital union (married/cohabiting). Other results are as shown in Table 1.

## Obstetric characteristics of participants

Table 2 shows that majority of the participants had two to four pregnancies and deliveries. Moreover, 37.3 percent in the intervention group and 45.3 percent in the control group started their first antenatal visit between 1 and 12 weeks as recommended. The minimum age of the respondent being pregnant at first was 14 years and maximum was 38 years with their mean age and standard deviation being 20 years (3.4). Of all respondents, 90.7 percent in the intervention group and 76.0 percent in the control group, respectively, had four or more ANC visits as recommended with p less than 0.001.

## ANC service utilization

**Overall Percentages on antenatal care service utilization of the study participants.** As shown in Table 3, overall ANC service utilization among the two groups in the post-test was found to be 77.3 percent in the intervention against 57.7 percent in the control group.

**Table 3. Overall antenatal service utilization among the study participants n = 450.**

| Score | Intervention | Control |
|---|---|---|
| = 3 (Received adequate services) | 116 (77.3%) | 173 (57.7%) |
| < 3 (Received Inadequate services) | 34 (22.7%) | 127 (42.3%) |
| Total | 150 | 300 |

## Factors related to effect of interactive messaging alert system on ANC service utilization

Some factors, which showed significant relationship with ANC service utilization apart from the intervention were age (p<0.01) and education level [(p<0.05) Table 4]. Other factors did not show significant relationship as shown in Table 4.

## Effect of mobile health and other factors on ANC utilization

Effect of Interactive Mobile Health system and other factors on ANC utilization in the study participants are shown in Table 5. Logistic regression analysis was employed. Findings indicated that the association between mobile health system and ANC utilization was as displayed in Table 5 (AOR = 2.164, P<0.05, 95% CI = 1.351–3.466) when adjusted for other factors. Thus, participants in the intervention group were two times more likely to utilize ANC services than the control group (Table 5). Other factors, which showed association included age category that ranged from 20 to 34 years (AOR = 1.877, P<0.05, 95%CI = 1.015–3.469) and education level at college/university level (AOR = 4.105, P<0.05, 95%CI = 1.479–11.390). Other factors did not show any significant association with ANC utilization as shown in Table 5.

**Table 4. Factors that may influence on antenatal care service utilization N = 450.**

| VARIABLE | RECEIVED ADEQUATE SERVICES | | RECEIVED INADEQUATE SERVICE | | P-VALUE |
|---|---|---|---|---|---|
| | n | % | n | % | |
| **AGE** | | | | | |
| <20 years | 31 | 43.7 | 40 | 56.3 | 0.000 |
| 20–34 years | 234 | 70.1 | 100 | 29.9 | |
| ≥35 years | 24 | 53.3 | 21 | 46.7 | |
| **EDUCATION LEVEL** | | | | | |
| Primary school | 155 | 58.7 | 109 | 41.3 | 0.001 |
| Secondary school | 100 | 68.0 | 47 | 32.0 | |
| College/university | 34 | 87.2 | 5 | 12.8 | |
| **OCCUPATIONAL STATUS** | | | | | |
| Non-employed | 94 | 65.3 | 50 | 34.7 | 0.757 |
| Self-employed | 189 | 63.4 | 109 | 36.6 | |
| Employed | 6 | 75.0 | 2 | 25.0 | |
| **MARITAL STATUS** | | | | | |
| Not married | 58 | 59.8 | 39 | 40.2 | 0.304 |
| Married | 231 | 65.4 | 122 | 34.6 | |
| **GRAVIDITY** | | | | | |
| 1 | 105 | 61.0 | 67 | 39.0 | 0.162 |
| 2–4 | 168 | 67.7 | 80 | 32.3 | |
| ≥ 5 | 16 | 53.3 | 14 | 46.7 | |
| **PARITY** | | | | | |
| 1 | 114 | 58.8 | 80 | 41.2 | 0.051 |
| 2–4 | 162 | 69.5 | 71 | 30.5 | |
| ≥ 5 | 13 | 56.5 | 10 | 43.5 | |

Table 5. Effect of Mobile health and other factors on ANC utilization.

| VARIABLE | OR | P-VALUE | CI 95% | | AOR | P-VALUE | CI 95% | |
|---|---|---|---|---|---|---|---|---|
| | | | Low | Upp | | | Low | Upp |
| **GROUPS** | | | | | | | | |
| Intervention | 2.505 | 0.000 | 1.604 | 3.911 | 2.164 | 0.001 | 1.351 | 3.466 |
| Control (**Ref.**) | | | | | | | | |
| **Age** <20 years (**Ref.**) | | | | | | | | |
| 20–34 years | 3.019 | 0.000 | 1.788 | 5.100 | 1.877 | 0.045 | 1.015 | 3.469 |
| ≥35 years | 1.475 | 0.310 | 0.696 | 3.123 | 0.896 | 0.814 | 0.359 | 2.238 |
| **Education level** | | | | | | | | |
| Primary school (**Ref.**) | | | | | | | | |
| Secondary school | 1.496 | 0.063 | 0.979 | 2.287 | 1.358 | 0.195 | 0.855 | 2.157 |
| College/university | 4.782 | 0.002 | 1.812 | 12.617 | 4.105 | 0.007 | 1.479 | 11.390 |
| **PARITY** | | | | | | | | |
| 1 (**Ref.**) | | | | | | | | |
| 2–4 | 1.601 | 0.021 | 1.074 | 2.387 | 1.354 | 0.235 | 0.821 | 2.233 |
| ≥ 5 | 0.912 | 0.837 | 0.381 | 2.183 | 1.304 | 0.611 | 0.469 | 3.627 |

Key Ref. = Reference Group

## Discussion

Findings from the current study showed antenatal care service utilization to be higher to participants in the intervention group than the control group. The overall ANC service utilization in the post-test was found to be high in the intervention group compared to the control group. The findings support results from the study, which was conducted in Ghana on mobile health technology in improving maternal and child health service utilization whereby they found the mobile technology to be effective [16].

Besides, presented findings from this study match with those found by Lund S and his fellow [17] through the study done in Zanzibar on effect of SMS on skilled delivery and access to emergency health care [18]. In the cited study, their results established that ANC utilization improved by 91 percent and thus, it demonstrates that mobile health system promotes ANC utilization compared to the traditional ANC services offered in the country's health facilities.

The findings are also supported by the results of the study done in South Africa on interactive mobile messaging programme to promote safe motherhood and improve pregnancy outcome whereby there was observed a significant increase in utilization of ANC services to the participants of that program [19].

The findings of this study is also supported by the systematic review on the use of mHealth to Improve Usage of Antenatal Care, Postnatal Care, and Immunization, which showed also some evidence of effectiveness at changing behavior to improve antenatal care attendance, postnatal care attendance, or childhood immunization rates [1]

Moreover, the research team carried out a logistic regression on effect of interactive mobile health technology on ANC service utilization. There was a positive association whereby intervention group members were two times more likely to utilize ANC services than the control group members after controlling the effect of other factors such as age and education level. Also, findings from the current study led to an observed low number of pregnant women who started their first ANC visit within the first trimester as recommended. This finding is similar to results from the study by TDHS of 2015/2016, which revealed only 24 percent of pregnant women started ANC in the first trimester. The first ANC visit at first trimester is encouraged and recommended for early identification of pregnant related complication(s) and for frequent as well as proper management and follow-up and this finding is also supported by the study

 

which was done in Dodoma in 2016 that shows only 12.4% of the women came for their first antenatal booking during the first trimester. The initiation of the program of ANC services requires good knowledge and awareness of appropriate preventive measures for the general public to ensure positive health behavior changes and reproductive health seeking habits in both partners [2]. Also, results from this study disclosed that 90.7 percent of the participants in the intervention group and 76.0 percent in the control group had four or more visits. The findings is similar with the findings of the systematic review that showed a positive effect of mHealth interventions on improving 4 or more ANC visit utilizations among pregnant women in Low and Middle Income Countries [3]. But this is a bit different with results from the study by TDHS 2015/2016 whereby there were 51 percent participants as recommended [8].

Such differences could be contributed by criteria used in enrolment of the study participants in the current study whereby only pregnant women who started ANC visit below 20 weeks were involved in this study. Also, it could be due to the reminder text messages sent to pregnant women in the intervention group about their next visits.

Other factors, such as education level, had a significant effect and should be addressed. Participants with college or university education were more likely to utilize ANC service than those with primary education level. It might be due to their high ability to understand and interpret the text messages sent to them as disclosed in other studies in high resource settings [18].

The current study demonstrated that the interactive mobile health system was effective in improving antenatal care service utilization. Simple SMS mobile phone messaging is a technology that can be applied to assist women and their families to seek timely as well as appropriate medical help for routine and emergent obstetric and newborn care.

## Limitations of the study

The most frequent encountered limitations included cellular and internet network connectivity. Network connectivity problems delay timely delivery of messages and thus, it was a challenge at times such that registration of clients was difficult. All required pieces of information were recorded in a paper format and the client was later registered in the system when the cellular network was good.

Telephone maintenance was another challenge among participants as some of them reported their phones to be out of order or stolen, an aspect, which prevented continuity of flow of messages to such particular participants. Moreover, there was another problem when the phone was stolen such that there was the risk of spill- over effect (information contamination) from the intervention to control group to occur. However, this was taken care by selecting intervention group and control group from different areas/health facilities.

Participation in the current study may have been limited due to the cost of phone service. Phone credits were not granted to participants due to financial constraints. As a result, some of the participants were unable to text or call back for assistance because they could not afford buying phone vouchers. Similarly, only women with phones could participate in the study and thus, women with fewer resources may have been excluded from the anticipated study sample.

Participants in the intervention and control groups were taken from different health facilities, aiming at minimizing information contamination that could have affected the results. Also, that could be a limitation because the manner of treatment of participants by different groups of healthcare workers might affect their compliance to health care services. Such limitation was taken care by training health care workers of both groups from different health facilities on the type of services to be offered.

 

Also the study included only pregnant women who started their first ANC visit in their first trimester as per WHO recommendations (20); those who started late were excluded so this might have limited them from getting health related information

## Conclusion

This study has demonstrated the importance of interactive mobile health technology in improving antenatal care service utilization compared to conventional method. The interactive mobile method improved health care utilization, raised awareness and served as the best alternative tool to provide health education among pregnant women. Therefore, the consistent use of SMS technology to disseminate health information is a promising approach in the country's setting to improve monitoring of pregnant women and increase maternal health care service utilization. Furthermore, mobile messaging programmes that provide health information messages have shown success in improving health outcomes in areas of maternal and child health, policy makers and planners in health-related issues such as the Ministry of health and other stakeholders should integrate this system in the country's antenatal care clinics. The system has proved to be a potentially useful innovation in low-resourced countries whereby providing comprehensive care including health education seems to be difficult due to staff shortage.

## Supporting information

**S1 CONSORT Checklist. CONSORT 2010 Checklist.**
(DOC)

**S1 CONSORT Flow Diagram. CONSORT 2010 Flow Diagram.**
(DOC)

**S1 File. Ethical clearance.**
(PDF)

**S2 File. Protocol.**
(DOCX)

## Author Contributions

**Conceptualization:** Theresia J. Masoi, Stephen M. Kibusi.

**Data curation:** Theresia J. Masoi, Deogratius Bintabara.

**Formal analysis:** Theresia J. Masoi, Stephen M. Kibusi.

**Investigation:** Theresia J. Masoi, Stephen M. Kibusi, Athanase Lilungulu.

**Methodology:** Stephen M. Kibusi, Deogratius Bintabara.

**Project administration:** Theresia J. Masoi, Stephen M. Kibusi.

**Resources:** Theresia J. Masoi.

**Software:** Deogratius Bintabara.

**Supervision:** Stephen M. Kibusi, Athanase Lilungulu.

**Validation:** Athanase Lilungulu.

**Writing – original draft:** Theresia J. Masoi, Deogratius Bintabara.

**Writing – review & editing:** Theresia J. Masoi, Stephen M. Kibusi, Deogratius Bintabara, Athanase Lilungulu.

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
