## [Decision Letter · Decision Letter 0]

26 Sep 2022

PDIG-D-22-00096

The effectiveness of interactive mobile health technologies in improving antenatal care service utilization in Dodoma region, Tanzania: a quasi - experimental study

PLOS Digital Health

Dear Dr. Bintabara,

Thank you for submitting your manuscript to PLOS Digital Health. After careful consideration, we feel that it has merit but does not fully meet PLOS Digital Health's publication criteria as it currently stands. Therefore, we invite you to submit a revised version of the manuscript that addresses the points raised during the review process.

When revising the manuscript, please take into account the comments by the reviewers, with a specific focus on the description of the intervention content and discussion of the implications of the findings.

Please submit your revised manuscript within 60 days Nov 25 2022 11:59PM. If you will need more time than this to complete your revisions, please reply to this message or contact the journal office at digitalhealth@plos.org. Please include the following items when submitting your revised manuscript:

We look forward to receiving your revised manuscript.

Kind regards,

Laura M. König

Academic Editor

PLOS Digital Health

Journal Requirements:

1. Please provide your detailed Financial Disclosure statement. This is published with the article. It must therefore be completed in full sentences and contain the exact wording you wish to be published.

a. Please clarify all sources of funding (financial or material support) for your study. List the grants (with grant number) or organizations (with url) that supported your study, including funding received from your institution. 

b. State the initials, alongside each funding source, of each author to receive each grant.

c. State what role the funders took in the study. If the funders had no role in your study, please state: “The funders had no role in study design, data collection and analysis, decision to publish, or preparation of the manuscript.”

d. If any authors received a salary from any of your funders, please state which authors and which funders.

2. Please provide separate figure files in .tif or .eps format only and remove any figures embedded in your manuscript file. Please also ensure that all files are under our size limit of 10MB.

4. In the online submission form, you indicated that "Data contain sensitive patient information, therefore, will be shared available upon request". All PLOS journals now require all data underlying the findings described in their manuscript to be freely available to other researchers, either 1. In a public repository, 2. Within the manuscript itself, or 3. Uploaded as supplementary information.

Additional Editor Comments (if provided):

1. Compared to the figures listed in the introduction, the recruited sample does not seem to be representative for the population, given that only participants could be enrolled who used ANC services early. Why did the authors decide for this criterion? This limitation should be reflected on in the discussion.

2. Please provide the formula used for estimating the required sample size.

3. Please provide more information about the intervention, especially on the intervention messages: What messages were sent? How were they developed? Did they include any specific intervention techniques? Were any aspects of the intervention derived from theory?

4. The differences in obstetric characteristics suggest that random allocation was unsuccessful. Please provide more information on the randomisation procedure that might explain why it failed (the authors might want to refer to the Cochrane Risk of Bias tool v2 to provide all information required for judging study quality) and discuss potential reasons and implications for this imbalance in the discussion.

5. The discussion describes additional results from a t-test. All results should be provided in the results section only, and be interpreted and discussed in the discussion section.

Reviewers' comments:

Reviewer's Responses to Questions

**Comments to the Author**

1. Does this manuscript meet PLOS Digital Health’s publication criteria? Is the manuscript technically sound, and do the data support the conclusions? The manuscript must describe methodologically and ethically rigorous research with conclusions that are appropriately drawn based on the data presented.

Reviewer #1: Yes

Reviewer #2: No

2. Has the statistical analysis been performed appropriately and rigorously?

Reviewer #1: Yes

Reviewer #2: I don't know

3. Have the authors made all data underlying the findings in their manuscript fully available (please refer to the Data Availability Statement at the start of the manuscript PDF file)?

Reviewer #1: No

Reviewer #2: Yes

4. Is the manuscript presented in an intelligible fashion and written in standard English?

Reviewer #1: Yes

Reviewer #2: No

5. Review Comments to the Author

Reviewer #1: The study assessed the effectiveness of an interactive mobile health technology in improving antenatal care service utilization in Tanzania. The study found that interactive mobile health technology if effective in improving antenatal care utilization.

Comments:

The paper is well written and easy to follow. However, the authors have used the phrase "in due regard" which i think can be replace with "in this regard". Please consult language editor.

Introduction:

Last sentence: This is where you are stating the aim of the study - (this study assessed efficacy of interactive mobile health technologies ...), this does not much with your title and the rest of the study. I think the right word is effectiveness not efficacy.

Methodology, results and discussion:

Methodology and results are well explained. 

Discussion need to be improved. You have discussed your results based of four references. I think you need to engage more with literature.

Conclusion.

In terms or novelty, this study offers none. Thus, you can bring in recommendations that can inform mobile health designers, implementers and policy makers. Here you need to think out of the box because this is the contribution of the study and it will make the study stronger.

Reviewer #2: Thank you for the opportunity to review the paper. While the intervention is potentially interesting, the description of the study, the methods and the analysis should be strengthened. It may be useful to review published papers which conduct bivariate and multivariate analysis to see how they explain some of the different steps. I also think some of the references need to be reviewed. Global recommendations since 2016 recommend 8 antenatal care contacts. I understand Tanzania still recommends four, so reference should be made to the Tanzania national norms.

6. PLOS authors have the option to publish the peer review history of their article (what does this mean?). If published, this will include your full peer review and any attached files.

**Do you want your identity to be public for this peer review?** For information about this choice, including consent withdrawal, please see our Privacy Policy.

Reviewer #1: No

Reviewer #2: No

---

## [Decision Letter · Decision Letter 1]

24 Feb 2023

PDIG-D-22-00096R1

The effectiveness of interactive mobile health technologies in improving antenatal care service utilization in Dodoma region, Tanzania: a quasi - experimental study

PLOS Digital Health

Dear Dr. Bintabara,

Thank you for submitting your manuscript to PLOS Digital Health. After careful consideration, we feel that it has merit but does not fully meet PLOS Digital Health's publication criteria as it currently stands. Therefore, we invite you to submit a revised version of the manuscript that addresses the points raised during the review process.

Please submit your revised manuscript within 30 days Mar 26 2023 11:59PM. If you will need more time than this to complete your revisions, please reply to this message or contact the journal office at digitalhealth@plos.org. Please include the following items when submitting your revised manuscript:

We look forward to receiving your revised manuscript.

Kind regards,

Laura M. König

Academic Editor

PLOS Digital Health

Journal Requirements:

Additional Editor Comments (if provided):

Unfortunately, some of the reviewers' concerns have not been adequately addressed. Specifically:

1) Please extend the discussions by additional references to previous research - it still includes only a very small number of references and so seems to ignore important parts of the literature

2) Please describe the intervention in more detail. For instance, what tips did women receive?

Reviewers' comments:

Reviewer's Responses to Questions

**Comments to the Author**

1. If the authors have adequately addressed your comments raised in a previous round of review and you feel that this manuscript is now acceptable for publication, you may indicate that here to bypass the “Comments to the Author” section, enter your conflict of interest statement in the “Confidential to Editor” section, and submit your "Accept" recommendation.

Reviewer #1: All comments have been addressed

2. Does this manuscript meet PLOS Digital Health’s publication criteria? Is the manuscript technically sound, and do the data support the conclusions? The manuscript must describe methodologically and ethically rigorous research with conclusions that are appropriately drawn based on the data presented.

Reviewer #1: Yes

3. Has the statistical analysis been performed appropriately and rigorously?

Reviewer #1: Yes

4. Have the authors made all data underlying the findings in their manuscript fully available (please refer to the Data Availability Statement at the start of the manuscript PDF file)?

Reviewer #1: No

5. Is the manuscript presented in an intelligible fashion and written in standard English?

Reviewer #1: Yes

6. Review Comments to the Author

Reviewer #1: Please use the language editor to proofread the manuscript. Check page 11 line 16 -19. Line 16 the sentence is not complete. And the next sentence is too long.

7. PLOS authors have the option to publish the peer review history of their article (what does this mean?). If published, this will include your full peer review and any attached files.

**Do you want your identity to be public for this peer review?** For information about this choice, including consent withdrawal, please see our Privacy Policy. 

Reviewer #1: No

---

## [Editor Report · Decision Letter 2]

10 Jul 2023

The effectiveness of interactive mobile health technologies in improving antenatal care service utilization in Dodoma region, Tanzania: a quasi - experimental study

PDIG-D-22-00096R2

Dear Dr Bintabara,

We are pleased to inform you that your manuscript 'The effectiveness of interactive mobile health technologies in improving antenatal care service utilization in Dodoma region, Tanzania: a quasi - experimental study' has been provisionally accepted for publication in PLOS Digital Health.

Best regards,

Laura M. König

Academic Editor

PLOS Digital Health